

# The value of diffusion weighted imaging-alberta stroke program early CT score in predicting stroke-associated pneumonia in patients with acute cerebral infarction: a retrospective study

Dong Zhao[1], Jing Zhu[1], Qiang Cai[2], Feifei Zeng[3], Xiujuan Fu[4] and Ke Hu[1]

[1] Department of Respiratory and Critical Care Medicine, Renmin Hospital of Wuhan University, Wuhan, China
[2] Department of Neurosurgery, Renmin Hospital of Wuhan University, Wuhan, China
[3] Department of Radiology, Renmin Hospital of Wuhan University, Wuhan, China
[4] Department of Neurology, Renmin Hospital of Wuhan University, Wuhan, China

Corresponding author
Ke Hu, huke-rmhospital@163.com

## ABSTRACT

**Background**. In this study, we aimed to investigate the value of Diffusion-Weighted Imaging-Alberta Stroke Program Early CT Score (DWI-ASPECTS) in predicting stroke-associated pneumonia (SAP) in patients with acute ischemic stroke.

**Methods**. A total of 291 patients who suffered acute cerebral infarction for the first time were included in this retrospective study. DWI-ASPECTS was assessed and clinical data were collected in order to find the risk factors of SAP, and a logistic regression model was used to investigate the effect of predicting SAP. Furthermore, correlation analysis was used to explore the relationship between DWI-ASPECTS and the immume status of the body.

**Results**. Among the 291 patients, 74 (25.4%) subjects were diagnosed with SAP. Compared with non-SAP, the patients with SAP were older and had a higher rate of atrial fibrillation (AF), National Institutes of Health Stroke Scale (NIHSS) scores. The SAP group also had a significantly lower DWI-ASPECTS than did the non-SAP group ($P < 0.01$). In the multivariable logistic regression analysis, the DWI-ASPECTS (adjusted odds ratio [aOR] = 1.438; 95% CI [1.158–1.787]; $P < 0.01$) remained significant after adjusting for confounders. What's more, the predictive ability of DWI-ASPECTS (AUC = 0.743 >0.7, 95% CI [0.678–0.800]) had acceptable discriminatory abilities. By the correlation analysis, DWI-ASPECTS was found to be negatively correlated with the count of white blood cell, neutrophils, monocytes, neutrophil-to-monocyte ratio and neutrophil-to-lymphocyte ratio, and positively correlated with the count of lymphocytes.

**Conclusions**. DWI-ASPECTS grades could predict stroke-associated pneumonia for patients with acute ischemic stroke, and combining grade with age, AF, or NIHSS could predict SAP events more accurately.

## INTRODUCTION

Stroke-associated pneumonia (SAP) is one of the most frequent complications following stroke, and it affects about 7% to 31.3% of all stroke patients (*Chinese Expert Consensus Group on Diagnosis and Treatment of Stroke-associated Pneumonia, 2010*; *Chumbler et al., 2010*; *Harms et al., 2013*; *Nam et al., 2018*; *Singer et al., 2009*). Evidence shows that SAP has a bad impact on patient outcome, lengthens the duration of hospitalization, and even increases long-term disability or mortality (*Alberti et al., 2011*; *Hilker et al., 2003*). There are several risk factors that relate to the development of pneumonia after stroke, all of which can be roughly grouped into two categories: the basic condition of the patient (*e.g.*, age, past medical history) and the stroke incident itself (*e.g.*, stroke severity or deficits such as conscious disturbance, dysphagia, and immunosuppression induced by stroke). Although there has been some progress in the treatment of SAP, early identification of patients of stroke with high risk of pneumonia might help to provide preventive measures and reduce the incidence of SAP (*Kalra et al., 2015*; *Meisel & Smith, 2015*; *Westendorp et al., 2015*).

Due to the lack of special clinical manifestations for the SAP in early stage, the pneumonia of these patients tends to be more serious when the SAP is diagnosed, which brings great challenges with regard to treatment. Several risk-scoring models have been proposed to help in the identification of patients at high risk of SAP (*Harms et al., 2013*; *Hoffmann et al., 2012*; *Ji et al., 2013*; *Kwon et al., 2006*). These predictive models are mostly based on clinical findings, but the clinical manifestations of SAP are frequently atypical and vague (*Kishore et al., 2015*; *Li et al., 2014*). Diffusion-Weighted Imaging-Alberta Stroke Program Early CT Score (DWI-ASPECTS), which comes from Alberta Stroke Program Early CT Score (ASPECTS), is simple and reliable and identifies stroke patients who are unlikely to make an independent recovery despite thrombolytic treatment (*Barber et al., 2000*; *Morita et al., 2009*). Researchers have adopted ASPECTS to evaluate the relationship between stroke localization and stroke-associated infection (SAI) (*Morita et al., 2009*). However, there is a lack of further study to confirm this value of predicting stroke-associated pneumonia for patients with acute ischemic stroke (AIS). The aim of this study was to assess the value of DWI-ASPECTS in predicting SAP for patients with AIS.

## MATERIALS & METHODS

### Patients and population

The retrospective study identified AIS patients ($n = 809$) who suffered stroke for the first time in Renmin Hospital of Wuhan University between August 2017 and August 2019. We enrolled 291 AIS patients based on the inclusion criteria, which were: patients with AIS who were admitted to hospital within 3 days of onset, AIS diagnosed based on the WHO criteria, and AIS confirmed by brain magnetic resonance image (MRI). The exclusion criteria were: a length of hospital stay <72 h or the onset time of AIS >72 h before admission ($n = 243$); diseases such as severe liver and kidney dysfunction, heart failure, tumour, and blood system and autoimmune diseases ($n = 64$); signs, symptoms, or reported suffering, at the time of admission, of chronic obstructive lung disease (COPD), asthma, or other pulmonary diseases ($n = 61$); a lack of diffusion-weighted image (DWI)

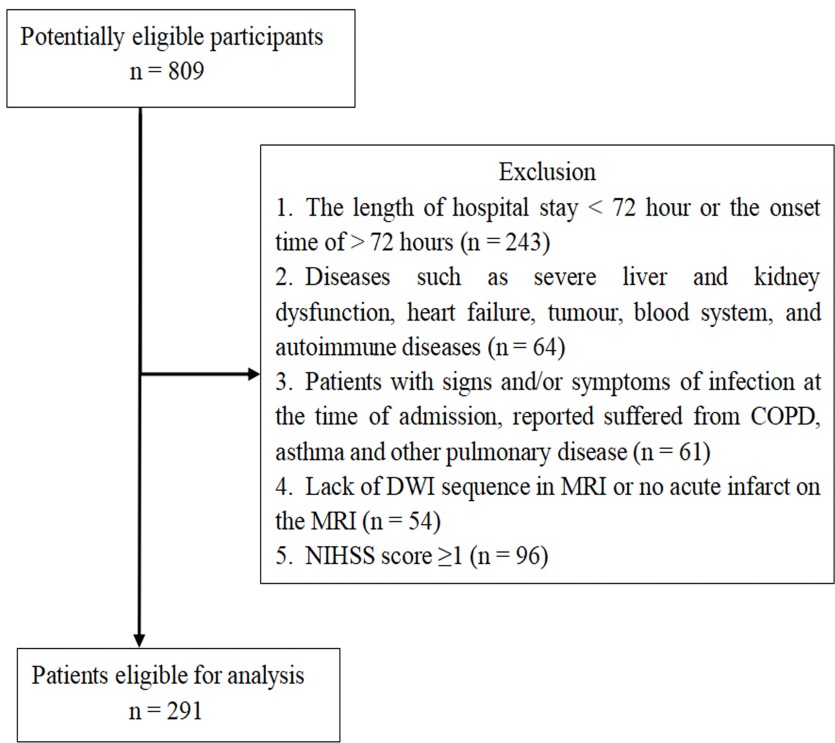

**Figure 1** **Patient Flow-Chart.** COPD = Chronic Obstructive Pulmonary Disease; MRI = Magnetic Resonance Imaging; DWI = Diffusion-Weighted Imaging; NIHSS = National Institutes of Health Stroke Scale; TIA = Transient Ischemic Attack.

sequence in MRI or no acute infarct on the MRI ($n = 54$); National Institutes of Health Stroke Scale (NIHSS) score = 0 ($n = 96$) (Fig. 1).

## Data collection and clinical assessment

The study protocol was approved by the ethics committee of Renmin Hospital of Wuhan University (2017K-C043). Informed consent, written or verbal, was obtained from all participants. The clinical data collection included prior medical history (hypertension, diabetes mellitus, and atrial fibrillation), result of routine blood test, partial blood biochemical examination (levels of glucose, lipids, albumin, homocysteine, C-reactive protein, serum amyloid A), head MRI, chest X-ray photographs or pulmonary CT, and NIHSS at admission. The neutrophil-to-lymphocyte ratio (NLR), known as a systemic inflammation and infection mark, has shown an excellent power for predicting pneumonia (*Curbelo et al., 2017*; *Lee et al., 2016*; *Nam et al., 2019*). Thus, we also evaluated the impact of NLR, as well as neutrophil-to-monocyte ratio (NMR), on patients with SAP. The NLR and NMR were calculated by dividing the absolute neutrophil count by the absolute lymphocyte and monocyte counts, respectively.

In this study, SAP was diagnosed according to the criteria of the Chinese Expert Consensus on Diagnosis and Treatment of Stroke-Associated Pneumonia (*Chinese Expert Consensus Group on Diagnosis and Treatment of Stroke-associated Pneumonia, 2010*). It

included newly emerging or progressively infiltrating pulmonary lesions in AIS patients' chest images combined with more than two of the following clinical manifestations of infection: (1) body temperatures exceeding 38 °C (rule out other causes of fever); (2) new respiratory symptoms such as cough and sputum, or pre-existing respiratory disease symptoms with or without chest pain or respiratory rate more than 25/min; (3) signs of pulmonary consolidation (percussion test presented with dull sound, enhanced vocal fremitus in palpation and/or moist rales in auscultation; (4) Peripheral blood white blood cell count $\geq 10 \times 10^9$/L or $\leq 4 \times 10^9$/L with or without left shift of nucleus. Some pulmonary diseases (tuberculosis, pulmonary tumour, non-infective interstitial lung disease, pulmonary oedema, pulmonary embolism, and pulmonary atelectasis), which might be similar to pneumonia with respect to clinical manifestations, were excluded.

Patients with acute ischemic stroke were arranged to complete a MRI examination within 24 h of their admission. The DWI-ASPECTS score was calculated at the start of the study based on the scoring rules which described in this manuscript. For anterior circulation stroke, we referred to the assessment method of Singer: the score calculated by the 10 regions shown in Fig. 2 (*Singer et al., 2009*). We subtracted 1 point for the area of early ischemic change for each of the defined regions. For posterior circulation stroke, we referred to the assessment method of Puetz , following the rule of subtracting 1 point for the regions of left or right thalamus, cerebellum, or PCA-territory, respectively; and subtracting 2 points for the regions of any part of the midbrain or pons (*Puetz et al., 2008*). A DWI-ASPECTS of 10 points meant no early ischemic change shown in the DWI scan; a 0 score implied ischemic involvement throughout the anterior circulation or the posterior circulation territory.

## Statistical analysis

Data were analysed using SPSS statistics 22.0 for windows (SPSS Inc., Chicago, IL). The measurement data with normal distribution were presented as mean $\pm$ standard deviation (SD), and median with interquartile range (IQR) were used for non-normal distribution. Continuous variables were analysed with Student's $t$-test or Mann–Whitney U-test, and categorical variables were analysed with Chi-square tests.

Factors with $P < 0.10$ in the univariate analysis were entered into the multivariate analysis. Variables of neutrophil and monocyte, NLR and MNR, WBC were entered into three multivariate logistic regression models separately, because of the high correlation with each other in univariate analysis.

Receiver operating characteristic (ROC) curve analyses were performed using MedCalc Version 15.8.0.0 (Frank Schoonjans, Mariakerke, Belgium). Accuracy in predicting outcome measures was assessed by calculating the area under ROC curve. Comparison of the areas under ROC curves (AUC) was performed using MedCalc Version 15.8.0.0. Correlations were analyzed using Spearman's correlations. All hypotheses were 2-tailed, and a $P$-value of $< 0.05$ was considered significant.

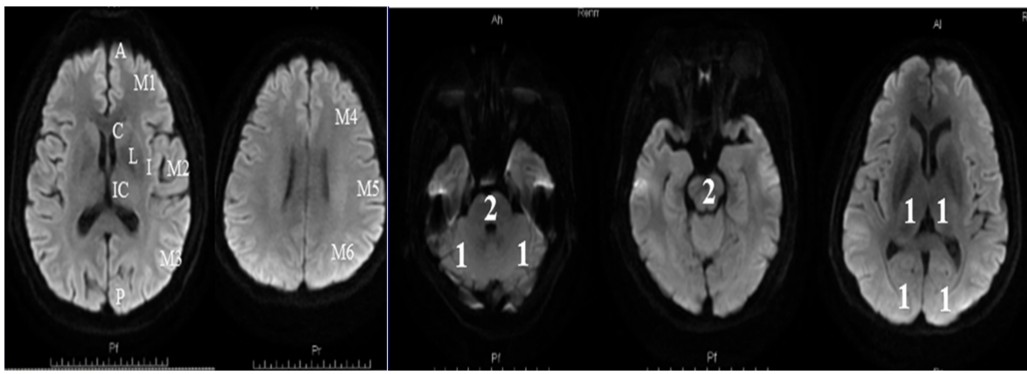

**Figure 2 Diffusion-Weighted Imaging-Alberta Stroke Program Early CT Score (DWI-ASPECTS).**
(A) The anterior circulation stroke assessed by DWI-ASPECTS. C = caudate, I = insular ribbon, L = lentiform, IC = internal capsule, M1 = anterior portion of the MCA cortex, M2 = MCA cortex lateral to the insular ribbon, M3 = posterior MCA cortex, M4 = anterior MCA territories, M5 = lateral MCA territories, and M6 = posterior MCA territories. One point was subtracted for the area of early ischemic change for each of the defined regions; (B) The posterior circulation stroke assessed by DWI-ASPECTS. From 10 points, 1 or 2 points each (as indicated) were subtracted for early ischemic changes in the left or right thalamus, cerebellum, or PCA territory, respectively (1 point); any part of midbrain or pons (2 points).

# RESULTS

Among the 291 patients, 74 (25.4%) subjects were diagnosed with SAP. The baseline characteristics between groups with and without SAP are presented in Table 1. An older age and a higher rate of atrial fibrillation (AF) presented in the SAP group. The SAP group also had higher initial NIHSS scores, WBC, neutrophil counts, levels of CRP, and SAA, while lower platelet (PLT) counts and levels of albumin (ALB) than those in the non-SAP group. What's more, the SAP group had a significantly lower DWI-ASPECTS than the non-SAP group (5 [3–7] *versus* 7 [6–8]; $P < 0.01$). In the SAP group, 15 patients had posterior circulation infarction and 59 anterior circulation infarction (Table 2). There was no significant difference in NIHSS between the groups of posterior circulation infarction and anterior circulation infarction.

The results of the multivariable logistic regression analysis was presented in Table 3. After adjusting for confounders, the difference of DWI-ASPECTS (adjusted odds ratio [aOR] = 1.438; 95% confidence interval [CI], 1.158–1.787; $P < 0.01$) remained significant. Furthermore, age (aOR = 0.950; 95% CI [0.918–0.984]; $P < 0.01$), AF (aOR = 0.193; 95% CI [0.071–0.526]; $P < 0.01$), and NIHSS score (aOR = 0.891; 95% CI [0.830–0.956]; $P < 0.01$) were also significant, aside from the altered DWI-ASPECTS. The results remained identical after replacing WBC or NLR and NMR with the neutrophil and lymphocyte counts as a sensitivity analysis.

From the ROC analysis, we found that the predictive ability of DWI-ASPECTS (AUC = 0.743 >0.7, 95% CI [0.678–0.800]) had acceptable discriminatory abilities. The optimal cut-off value was 6 for DWI-ASPECTS, whose sensitivity was 74.32%, specificity was 70.51%, positive predictive value (PPV) was 46.22%, and negative predictive value (NPV)

**Table 1  Baseline characteristics of the patients.**

| Characteristics[a] | SAP ($n = 74$) | Non-SAP ($n = 217$) | P -Value |
|---|---|---|---|
| Male: Female[§] | 47:27 | 142:75 | 0.764 |
| Age, years[‡] | 72.32 ± 13.67 | 64.19 ± 12.67 | 0.000 |
| NIHSS[†] | 12 (5–16) | 4 (2–7) | 0.000 |
| DWI-ASPECTS[†] | 5 (3–7) | 7 (6–8) | 0.000 |
| Hypertension, n (%)[§] | 47 (63.5) | 137 (63.1) | 0.953 |
| Diabetes, n (%)[§] | 17 (23.0) | 71 (32.7) | 0.115 |
| Atrial fibrillation, n (%)[§] | 17 (23.0) | 22 (10.1) | 0.005 |
| Neutrophil ($\times 10^9$/L)[†] | 6.08 (4.01–9.03) | 4.72 (3.54–6.32) | 0.001 |
| Monocyte ($\times 10^9$/L)[†] | 0.55 (0.42–0.73) | 0.51 (0.41–0.68) | 0.298 |
| Lymphocyte ($\times 10^9$/L)[†] | 1.43 (0.85–1.79) | 1.55 (1.22–2.02) | 0.017 |
| WBC ($\times 10^9$/L)[‡] | 8.70 ± 3.25 | 7.50 ± 2.36 | 0.005 |
| RBC ($\times 10^{12}$/L)[‡] | 4.47 ± 0.68 | 4.59 ± 0.62 | 0.159 |
| PLT ($\times 10^9$/L)[‡] | 193.21 ± 75.00 | 214.64 ± 66.93 | 0.024 |
| SAA (mg/L)[†] | 10.06 (5–52.51) | 5.07 (5.00–9.76) | 0.002 |
| CRP (mg/L)[†] | 5.06 (1.62–15.02) | 2.13 (0.73–5.86) | 0.001 |
| Albumin (g/L)[‡] | 39.97 ± 4.65 | 41.49 ± 3.75 | 0.006 |
| Blood glucose (mmol/L)[†] | 6.29 (5.20–7.80) | 5.64 (4.80–7.21) | 0.076 |
| Triglyceride (mmol/L)[†] | 1.2 (0.87–1.73) | 1.37 (1.03–2.03) | 0.079 |
| High-density lipoprotein (mmol/L )[†] | 1.1 (0.88–1.39) | 1.03 (0.85–1.25) | 0.054 |
| Low-density lipoprotein (mmol/L)[†] | 2.28 (1.82–3.06) | 2.55 (2.05–3.08) | 0.123 |
| HCY ($\mu$mol/L)[†] | 17.33 (12.74–22.60) | 15.41 (12.91–19.65) | 0.487 |
| Ischemia region[§] | | | 0.532 |
| Anterior circulation, n (%) | 59 (24.70) | 180 (75.30) | |
| Posterior circulation, n (%) | 15 (28.80) | 37 (71.20) | |
| Brain stem infarcts, n (%)[§] | 11 (14.9) | 24 (11.1) | 0.385 |

**Notes.**

[a] Continuous variables were expressed as mean ± standard deviation or as median (interquartile range). Categorical variables were expressed as frequency (percent).

[†] Mann–Whitney U-test.

[‡] Student's t-test.

[§] $\chi^2$ test.

Abbreviations: NIHSS, National Institutes of Health Stroke Scale; DWI-ASPECTS, Diffusion Weighted Imaging-Alberta Stroke Program Early CT Score; WBC, White Blood cell Count; RBC, Red Blood cell Count; PLT, Platelet Count; SAA, Serum Amyloid A; CRP, C-Reactive Protein; HCY, Homocysteine.

**Table 2  The characteristics of different ischemic region in SAP patients.**

| | Anterior circulation ($n = 59$) | Posterior circulation ($n = 15$) | P |
|---|---|---|---|
| NIHSS[†] | 13 (7–16) | 8 (3–20) | 0.215 |

**Notes.**

[†] Continuous variables were expressed as media (interquartile range). Mann-Whitney U-test was used for statistical analysis.

was 88.95%. Other independent risk factors (age, NIHSS, and AF) of SAP in our study were analyzed to investigate the different powers of predictive performance (Fig. 3). Although NIHSS (AUC = 0.778, 95% CI [0.716–0.832]) showed the highest AUC out of all the independent risk factors referred above, there was no significant difference between

**Table 3  Logistic regression of associations between risk factors for SAP.**

| Parameters | Model 1 | | | | | |
|---|---|---|---|---|---|---|
| | B | SE | Wald | P | OR | 95% CI |
| Sex | −0.231 | 0.444 | 0.271 | 0.603 | 0.794 | 0.332–1.895 |
| Age | −0.051 | 0.018 | 8.259 | 0.004 | 0.950 | 0.918–0.984 |
| DWI-ASPECTS | 0.363 | 0.111 | 10.765 | 0.001 | 1.438 | 1.158–1.787 |
| NIHSS | −0.116 | 0.036 | 10.269 | 0.001 | 0.891 | 0.830–0.956 |
| AF | −1.647 | 0.512 | 10.337 | 0.001 | 0.193 | 0.071–0.526 |
| [1] Neutrophil | −0.053 | 0.089 | 0.359 | 0.549 | 0.948 | 0.797–1.128 |
| [1] Lymphocyte | 0.157 | 0.339 | 0.214 | 0.644 | 1.170 | 0.602–2.271 |
| [2] NLR | −0.009 | 0.036 | 0.066 | 0.797 | 0.991 | 0.922–1.064 |
| [2] NMR | −0.021 | 0.061 | 0.117 | 0.733 | 0.980 | 0.870–1.103 |
| [3] WBC | −0.030 | 0.087 | 0.118 | 0.731 | 0.971 | 0.818–1.151 |
| PLT | 0.001 | 0.003 | 0.106 | 0.745 | 1.001 | 0.995–1.007 |
| SAA | −0.003 | 0.004 | 0.609 | 0.435 | 0.997 | 0.990–1.004 |
| CRP | −0.017 | 0.011 | 2.629 | 0.105 | 0.983 | 0.962–1.004 |
| ALB | 0.094 | 0.051 | 3.467 | 0.063 | 1.099 | 0.995–1.214 |
| Blood sugar | −0.053 | 0.069 | 0.579 | 0.447 | 0.949 | 0.828–1.087 |
| Triglyceride | 0.055 | 0.230 | 0.058 | 0.810 | 1.057 | 0.673–1.660 |
| HDL | −0.942 | 0.580 | 2.643 | 0.104 | 0.390 | 0.125–1.214 |

**Notes.**

Abbreviations: AF, Atrial fibrillation; HDL, High-density lipoprotein; NLR, neutrophil-tolymphocyte ratio; NMR, neutrophil-to-monocyte ratio.

We brought sex which reported a risk factor of SAP into analysis, although it was shown to have no significance in this study.

Model 1: Adjusted with $P < 0.10$ in univariate analysis (Sex, Age, DWI-ASPECTS, NIHSS, AF, Neutrophil, Lymphocyte, PLT, SAA, CRP, ALB, and HDL).

Model 2: Adjusted with $P < 0.10$ in univariate analysis (Sex, Age, DWI-ASPECTS, NIHSS, AF, NLR, NMR, PLT, SAA, CRP, ALB, and HDL).

Model 3: Adjusted with $P < 0.10$ in univariate analysis (Sex, Age, DWI-ASPECTS, NIHSS, AF, WBC, PLT, SAA, CRP, ALB, and HDL).

NIHSS and DWI-ASPECTS ($P > 0.05$). We further calculated the AUC which combined DWI-ASPECTS, age, NIHSS, and AF, and found that the AUC was significantly greater than age, NIHSS, AF, and DWI-ASPECTS alone for both outcomes (all $p < 0.01$) in either case (Table 4).

This study further explored the correlation between DWI-ASPECTS and the immune status of AIS patients. Although the correlation was not strong, there was significant negative correlation with WBC ($r = −0.21$, $p < 0.01$), Neu ($r = −0.26$, $p < 0.01$), Mon ($r = −0.17$, $P < 0.05$), NLR ($r = −0.12$, $p < 0.05$), NMR ($r = −0.30$, $p < 0.01$) and SAA ($r = −0.18$, $p < 0.05$), and a significant positive correlation with lymphocyte counts ($r = 0.19$, $p < 0.01$) and PLT ($r = 0.18$, $P < 0.01$), shown in Fig. 4.

## DISCUSSION

Early and accurate diagnosis of SAP was regarded as necessary for timely and effective treatment. Originally, ASPECTS and posterior circulation Acute Stroke Prognosis Early CT Score (pc-ASPECTS) were used to identify the early ischemic change in the region of anterior circulation and posterior circulation, respectively. Harms employed ASPECTS

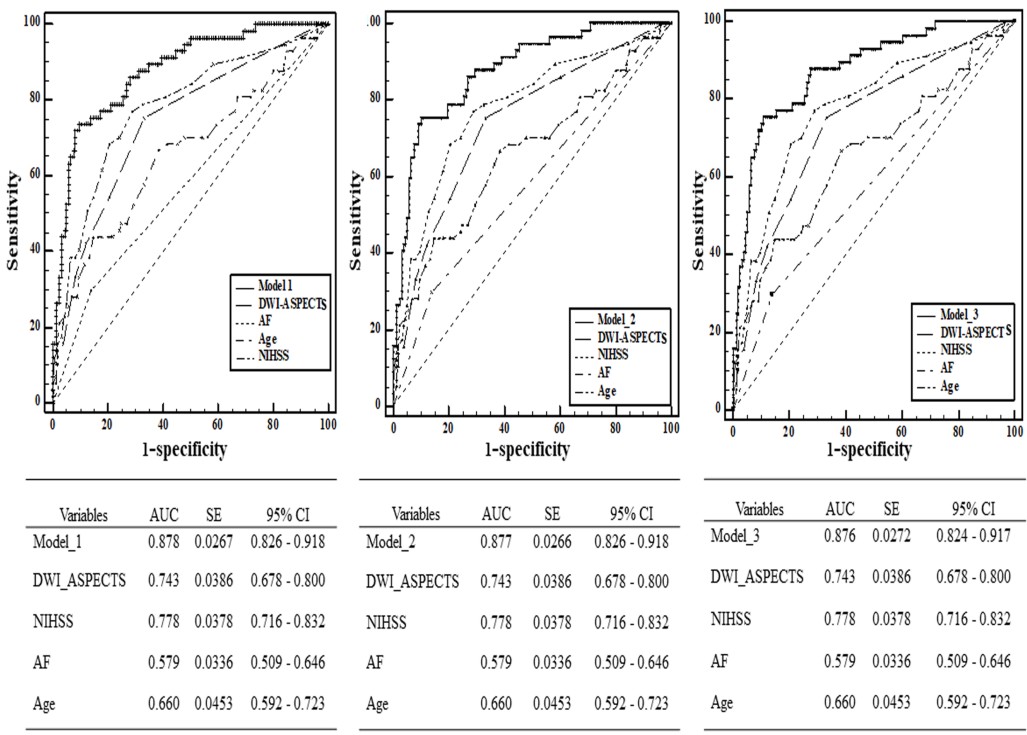

| Variables | AUC | SE | 95% CI |
|---|---|---|---|
| Model_1 | 0.878 | 0.0267 | 0.826 - 0.918 |
| DWI_ASPECTS | 0.743 | 0.0386 | 0.678 - 0.800 |
| NIHSS | 0.778 | 0.0378 | 0.716 - 0.832 |
| AF | 0.579 | 0.0336 | 0.509 - 0.646 |
| Age | 0.660 | 0.0453 | 0.592 - 0.723 |

| Variables | AUC | SE | 95% CI |
|---|---|---|---|
| Model_2 | 0.877 | 0.0266 | 0.826 - 0.918 |
| DWI_ASPECTS | 0.743 | 0.0386 | 0.678 - 0.800 |
| NIHSS | 0.778 | 0.0378 | 0.716 - 0.832 |
| AF | 0.579 | 0.0336 | 0.509 - 0.646 |
| Age | 0.660 | 0.0453 | 0.592 - 0.723 |

| Variables | AUC | SE | 95% CI |
|---|---|---|---|
| Model_3 | 0.876 | 0.0272 | 0.824 - 0.917 |
| DWI_ASPECTS | 0.743 | 0.0386 | 0.678 - 0.800 |
| NIHSS | 0.778 | 0.0378 | 0.716 - 0.832 |
| AF | 0.579 | 0.0336 | 0.509 - 0.646 |
| Age | 0.660 | 0.0453 | 0.592 - 0.723 |

**Figure 3 Comparison of area under curve (AUC) between DWI-ASPECTS and other independence factors in the prediction of SAP by Receiver Operating Characteristic (ROC) curves.** The additive effect of DWI-ASPECTS, AF, Age and NIHSS in the prediction of SAP was also calculated.

**Table 4 Pairwise comparison of ROC curves.**

| Parameters | | Difference between areas | Standard Error | 95% Confidence interval | Z statistic | P-Value |
|---|---|---|---|---|---|---|
| AF | model_1 | 0.299 | 0.0379 | 0.225–0.373 | 7.900 | 0.000 |
| | model_2 | 0.299 | 0.0378 | 0.225–0.373 | 7.912 | 0.000 |
| | model_3 | 0.297 | 0.0380 | 0.222–0.372 | 7.811 | 0.000 |
| Age | model_1 | 0.218 | 0.0447 | 0.130–0.306 | 4.872 | 0.000 |
| | model_2 | 0.218 | 0.0447 | 0.130–0.305 | 4.867 | 0.000 |
| | model_3 | 0.216 | 0.0448 | 0.128–0.304 | 4.815 | 0.000 |
| DWI_ASPECTS | model_1 | 0.135 | 0.0372 | 0.0621–0.208 | 3.629 | 0.000 |
| | model_2 | 0.135 | 0.0370 | 0.0622–0.207 | 3.640 | 0.000 |
| | model_3 | 0.133 | 0.0370 | 0.0604–0.206 | 3.592 | 0.000 |
| NIHSS | model_1 | 0.0995 | 0.0320 | 0.0368–0.162 | 3.111 | 0.002 |
| | model_2 | 0.0992 | 0.0320 | 0.0366–0.162 | 3.106 | 0.002 |
| | model_3 | 0.0974 | 0.0320 | 0.0347–0.160 | 3.044 | 0.002 |
| DWI_ASPECTS ~NIHSS | | 0.0356 | 0.0385 | −0.0398–0.111 | 0.925 | 0.355 |

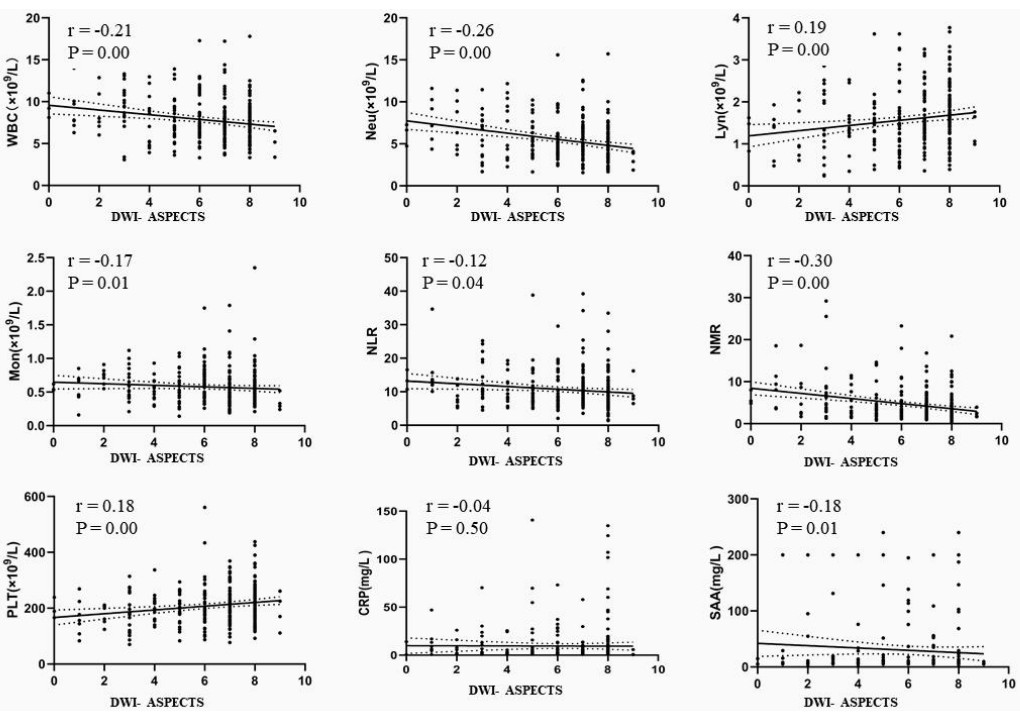

**Figure 4** **The correlation between DWI-ASPECTS and the immune status of patients with AIS.** AIS = acute ischemic stroke, WBC = white blood cell count, Neu = neutrophil count, Lyn = lymphocyte count, Mon = monocyte count, NLR = neutrophil-to-lymphocyte ratio, NMR = neutrophil-to-monocyte ratio, PLT = platelet count, SAA = serum amyloid A, CRP = C-reactive protein.

graded stroke localization to investigate ischemic lesion characteristics (size, localization) correlation with immune competence (monocytic human leukocyte antigen-DR [HLA-DR] expression) and post-stroke infections (*Harms et al., 2011*). Thus, considering it is more conclusive to find the stroke localization within a short time by DWI sequence in MRI (*Sugimori et al., 2012*), this study adopted DWI-ASPECTS to explore the relationship with SAP and test the value in predicting the risk of SAP for AIS patients. The results suggested that DWI-ASPECTS has a good predictive value for the diagnosis of SAP and shows an excellent predictive ability when combined with age, AF, and NIHSS.

NIHSS was a 15-item impairment scale which recommended as a valid tool to assess stroke severity (*Kasner, 2006*), and shown as good predictive value as DWI-ASPECTS for the diagnosis of SAP in this study. To reduce the detection bias, clinicians needed to be trained in how to accurate evaluation of NIHSS (*Lyden et al., 2005*). In contrast, the evaluation method of DWI-ASPECTS which calculated on the basis of special neuroanatomical area seemed to be more convenient. Assessors could follow the neuroanatomical areas to score objectively, when the DWI sequence in MRI was completed. However, it was undeniable that NIHSS has an important clinical application in predicting the outcome for stroke patient.

As we all know, there are various factors (consciousness disturbance, dysphagia, aspiration, over-activation of the sympathetic nerve system, stroke-induced immunodepression,

etc.) that contribute to the occurrence of SAP. What's more, several studies have reported that special anatomy of ischemic lesions and infarct volume, which is associated with systemic immunodepression, might contribute to the high risk of post-stroke infection as well (*Urra et al., 2017*). The semiquantitative scoring system of DWI-ASPECTS was calculated on the basis of special neuroanatomical area and reflected the volume of infarct to a certain degree. In the course of investigating the possible mechanism that DWI-ASPECTS might be a predictor of SAP, the magnitude of the DWI-ASPECTS was associated with peripheral immune changes by doing the correlation analysis between DWI-ASPECTS and peripheral blood inflammatory cells.

Stroke-induced immunosuppression is an important risk factor of SAP has been widely recognized (*Chamorro et al., 2012*; *Hoffmann et al., 2017*; *Kemmling et al., 2013*; *Mracsko et al., 2014*). A characteristic of stroke-induced immunosuppression is the impairment of immune function, such as a decrease in the number of circulating lymphocytes and deactivation of monocytes. Lymphopenia, in turn, may predispose to the development of infections (*Merayo-Chalico et al., 2013*; *Klehmet et al., 2009*). Compared with non-SAP patients in our study, the SAP patients had higher levels of WBC and neutrophil and lower levels of lymphocyte, which was in line with previous studies (*Dirnagl et al., 2007*). However, among these factors only DWI-ASPECTS, age, AF, and NIHSS remained significant contributors after adjusting for confounders in the multivariable logistic regression model. Surprisingly, to some extent, lymphopenia didn't remain as independent factor in the multivariable regression model. After consideration, we could give explanations as follows. High initial NIHSS score, DWI-ASPECTS or lymphopenia could be used to represent stroke severity. Lymphopenia not remaining significant was potentially due to interaction effects influenced by many substances and the surrounding microenvironments. In addition (*Feng et al., 2018*; *Nam et al., 2018*; *Westendorp et al., 2015*), a high NLR was reported to show a good correlation with SAP events in patients with AIS. As is shown in Fig. 4, these results indirectly suggest that DWI-ASPECTS has relation to immunosuppression, although the correlation was not strong. Of course, these assumptions need more experimental data to validate and more supports from clinical data that excluding the bias caused by the low patient case number.

Besides, aspiration which was likely caused by dysphagia was another independent risk factor of SAP (*Chamorro et al., 2012*; *Hoffmann et al., 2017*). It was reported that dysphagia carried a seven fold increased risk of pneumonia after stroke (*Singh & Hamdy, 2006*). The survivor of brainstem stroke might experience deficits in pharyngeal and upper sophageal sphincter function, and thus further increase the risk of SAP (*Prabhjot, 2016*; *Kaur, 2016*; *Sasegbon & Hamdy, 2017*). Thus, the management of dysphagia was one of the key issues in preventing the occurrence of aspiration pneumonia in stroke patients (*Radhakrishnan, Menon & Anandakuttan, 2013*). Compared with these did not reach brainstem, ischemic lesions reached brainstem in AIS patients exerted no impact on the development of SAP in this study. The probable reason is that more preventive care were given to these AIS patient with brainstem stroke.

There were several limitations to this study. Firstly, this study was a retrospective study. Although a relatively large sample size was involved, about one third of the AIS patients

who suffered stroke for the first time were included in the analysis. Therefore, a possibility of selection bias inevitably existed. We adopted the Chinese Expert Consensus on Diagnosis and Treatment of Stroke-Associated Pneumonia as diagnostic criteria for SAP as well as Li (*Li et al., 2014*), and calculated that the incidence rate of SAP was 25.4%, which was in accord with the result of Li (24.1%). The change in immune conditions between non-SAP and SAP patients concur with the opinion of stroke-induced immunosuppression. Taking these results, which align with previous research, into account, we believed that this research approach was reasonable and reliable. Secondly, the record of dysphagia screening was incomplete in this retrospective study, which might be a drawback. At the same time, there was a lack of specific recommendations on the standardized criteria for dysphagia screening in international stroke guidelines. Thirdly, DWI-ASPECTS based on the diffusion-weighted image of MRI was simpler and more sensitive than CT scan in finding the EIC, and it showed a strong predictive ability of SAP occurrence for AIS patients. But before generalization of the findings to clinical fields, a prospective, multicentre study was needed.

## CONCLUSIONS

In conclusion, DWI-ASPECTS has a good predictive value for the diagnosis of SAP. Because the DWI-ASPECTS can be easily calculated from the DWI sequence in MRI examination, it may help to select high-risk patients to begin intervention in time. However, to prove its validity further, DWI-ASPECTS needs to be assessed prospectively and at multiple centres.

### Funding

This work was supported by the National Natural Science Foundation of China grant (No. 81770089, 81970082). The funders had no role in study design, data collection and analysis, decision to publish, or preparation of the manuscript.

### Grant Disclosures

The following grant information was disclosed by the authors:
National Natural Science Foundation of China: 81770089, 81970082.

### Competing Interests

The authors declare there are no competing interests.

### Author Contributions

- Dong Zhao conceived and designed the experiments, performed the experiments, analyzed the data, prepared figures and/or tables, authored or reviewed drafts of the paper, and approved the final draft.
- Jing Zhu performed the experiments, analyzed the data, authored or reviewed drafts of the paper, and approved the final draft.
- Qiang Cai, Feifei Zeng and Xiujuan Fu performed the experiments, prepared figures and/or tables, and approved the final draft.

- Ke Hu conceived and designed the experiments, performed the experiments, authored or reviewed drafts of the paper, and approved the final draft.

## Human Ethics

The following information was supplied relating to ethical approvals (i.e., approving body and any reference numbers):

The study protocol was approved by the ethics committee of Renmin Hospital of Wuhan University (Ethical Application Ref:2017K-C043).

## Data Availability

The raw measurements are available in the Supplementary File.

## Supplemental Information

Supplemental information for this article can be found online at http://dx.doi.org/10.7717/peerj.12789#supplemental-information.

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
