# Peer review of "The value of diffusion weighted imaging-alberta stroke program early CT score in predicting stroke-associated pneumonia in patients with acute cerebral infarction: a retrospective study"

_PeerJ, doi:10.7717/peerj.12789_

## Round 0.1 · original submission · Major Revisions

Dear Authors,

There are major issues in your manuscript that need revision. Please read the comments of the two Peer Reviewers carefully and revise the manuscript accordingly.

Reviewer 1 ·

Basic reporting

The article has many grammatical errors (eg line 83, 255) and needs to be corrected.

Experimental design

1. How and when was the DWI ASPECTS score calculated ? More details to be given.
2. Exclusion criteria on the length of hospital stay is unclear. Line 108 and Figure 1 patient flow chart not consistent.
3. Line 218, NIHSS score of 15 items needs clarification as NIHSS score has only 11 items.

Validity of the findings

The size of the infarct (low DWI ASPECTS score) indicates the severity & the extent of the infarct and the moribund state of the patient's illness. Therefore, it is understandable that the larger the infarct the chances of getting any of the complications that can occur in stroke patients including SAP may be higher. There seems to be no direct causal relationship between the two parameters studied.
Many other parameters especially CRP, DM, AF etc., if abnormal, can in varying combinations may have contributed to the development of SAP.

Additional comments

The manuscript very vague and too long and the authors are not focussing on the message they want to convey.

Reviewer 2 ·

Basic reporting

See all comments below.

Experimental design

See all comments below.

Validity of the findings

See all comments below.

Additional comments

In the study, Zhao et al investigated the association between ASPECTS and occurrence of stroke-associated pneumonia. Even after controlling for multiple, relevant confounders, most notably NIHSS, ASPECTS was still predictive of SAP. The study design is logical, the statistics sound and the manuscript as a whole is easy to read.
Comments:
1. The study’s main finding probably only reflects that the risk of SAP increases with the extent of brain injury. It is somewhat unexpected that ASPECTS had a predictive value even when NIHSS was controlled for. This is possibly due to the fact that admission NIHSS, and not NIHSS at a later time-point, was used in the analysis. Admission NIHSS is reasonably less associated to the extent of permanent brain injury than NIHSS after a few days would be. I have no issue with the use of admission NIHSS, it is indeed the most common practice, but it could be the reason why so much explanatory value was left for ASPECTS.
2. Why have patients that stayed less than 72 h been excluded from the analysis? Isn’t it hard for the clinician to determine beforehand that the patient will leave early? This exclusion criterion limits the generalizability of the study.
3. When was the MRI performed, in relation to the stroke? If the MRI was performed later than 72 h, many patients will already have developed their SAP (ref: PMID: 8610305 och 10447789 och 12637700), which could bias the ASPECTS assessment.
4. The authors need to elaborate the SAP criteria. For example: Was pulmonary consolidations a radiological sign?
5. The authors point out that MRI may be more convenient than NIHSS, but clearly they cannot mean that NIHSS should be replaced by ASPECTS (lines 218-224)? The predictive value of NIHSS was also solid, which could be emphasized.
6. Row 225-242: This section contains a reasoning about how different localizations could affect different functions and lead to SAP. The section is a bit vague, and the fact is that the study doesn't try to evaluate how damage in different areas could affect the risk of SAP. The exception being that the authors note that anterior and posterior stroke has the same risk of SAP. They also note that posterior stroke has a lower DWI-ASPECT-score but since they have a different scoring system for anterior and posterior stroke, it is impossible to draw any conclusions from this. In row 253-258 they do note that brain-stem lesions (in this study) are not associated with increased risk of SAP, and while brain stem stroke often comes with dysphagia it is also common in other types of stroke. Since dysphagia wasn't recorded, the conclusion "Thus, the results revealed that dysphagia did not all of the reason for SAP." is hard to draw.
7. It is an interesting finding that while lymphopenia is associated with a large stroke (and thus higher risk of SAP), in the logistic regression it is not an independent predictor for SAP. My reasoning is that this would suggest that lymphopenia is hardly a major causal pathway by which stroke leads to SAP. Of course it may also be a result of too few subjects. Maybe they could include some comment on this?
8. Table 3 is not easily read, and the different models don't differ in their results. It would be better to just present one of the models with a comment that neutrophils, lymphocytes, NLR, NMR, WBC are all insignificant predictors regardless of model.
9. Further, table 4 needs a better explanatory text.
10. Abstract: “Immume” should be “immune”.
11. Line 108: The sentence "a length of hospital stay <72 hours or the onset time of AIS>72 hours " could be clarified. More than 72 hours before what? Admission? MRI?
12. Line 216: The sentence starting on this line seems to have been cut from the previous sentence. Rephrase.
13. Line 258: The sentence “Thus, the results revealed that dysphagia did not all of the reason for SAP.” Doesn’t make sense. Rephrase.
14. Line 132: Peculiar expression: "expectoration emerging"

---

## Round 0.2 · Minor Revisions

Please revise the manuscript as per comments:
Row 48: change ”immume” to ”immune”

The section in the Discussion starting with ” Aspiration and stroke-induced immunosuppression…”: Sorry, but the shortcomings of the language in this section are so pronounced that it is hard to understand what the authors are meaning. It needs some serious proof-reading. For example, the sentence ” It was reported that the swallowing center within the brainstem which controls and manipulates the entire pharynx phase swallowing sequence (Sasegbon A et al., 2017).” doesn’t make sense. (Also, ”esophagus” lacks an e.)

Reviewer 2 ·

Basic reporting

See below.

Experimental design

See below.

Validity of the findings

See below.

Additional comments

Row 48: change ”immume” to ”immune”

Section in Discussion starting with ” Aspiration and stroke-induced immunosuppression…”: Sorry, but the shortcomings of the language in this section are so pronounced that it is hard to understand what the authors are meaning. Needs som serious proof-reading. For example, the sentence ” It was reported that the swallowing center within the brainstem which controls and manipulates the entire pharynx phase swallowing sequence (Sasegbon A et al., 2017).” doesn’t make sense. (Also, ”esophagus” lacks an e.)

---

## Round 0.3 · accepted · Accept

Dear Authors, I have re-reviewed your revised manuscript and looked at the rebuttals, and decided to accept it. Thank You.